# CellCycleTRACER accounts for cell cycle and volume in mass cytometry data

Maria Anna Rapsomaniki[1], Xiao-Kang Lun [2,3], Stefan Woerner[1], Marco Laumanns[1,4], Bernd Bodenmiller [2] & María Rodríguez Martínez [1]

Recent studies have shown that cell cycle and cell volume are confounding factors when studying biological phenomena in single cells. Here we present a combined experimental and computational method, CellCycleTRACER, to account for these factors in mass cytometry data. CellCycleTRACER is applied to mass cytometry data collected on three different cell types during a TNFα stimulation time-course. CellCycleTRACER reveals signaling relationships and cell heterogeneity that were otherwise masked.

---

[1] Zürich Research Lab, IBM, Säumerstrasse 4, 8803 Rüschlikon, Switzerland. [2] Institute of Molecular Life Sciences, University of Zürich, Winterthurerstrasse 190, 8057 Zürich, Switzerland. [3] Molecular Life Science Ph.D. Program, Life Science Zürich Graduate School, ETH Zürich and University of Zürich, 8057 Zürich, Switzerland. [4] Present address: BestMile SA, EPFL Innovation Park, Building D, 1015 Lausanne, Switzerland. Maria Anna Rapsomaniki and Xiao-Kang Lun contributed equally to this work. Bernd Bodenmiller and María Rodríguez Martínez jointly supervised this work. Correspondence and requests for materials should be addressed to B.B. (email: bernd.bodenmiller@imls.uzh.ch) or to M.R.Mín. (email: mrm@zurich.ibm.com)

Single-cell analysis technologies are rapidly improving and will eventually match the performance of their population-level counterparts. RNA transcriptomes can be quantified in thousands of single cells, and analyses of transcriptomes of single cells with spatial resolution in tissues have been reported[1-3]. Mass cytometry has the potential to enable simultaneous detection of up to 50 proteins, protein modifications, such as phosphorylation, and transcripts in single cells[4-7]. Recent developments enable highly multiplexed imaging of similar numbers of markers in adherent cells and tissues[5,8,9,10].

Single-cell data are typically used to identify cell subpopulations that share similar transcript or protein expression or functional markers. Analyses of these subpopulations can be used to reveal differences between tissue compartments in health and disease[11-14], to reconstruct signaling network interactions, to study regulatory mechanisms[15-17], and, together with clinical data, to identify single-cell features that predict characteristics such as response to treatment and likelihood of relapse[18]. For continuous processes, such as stem cell differentiation and the cell cycle, single-cell data allow the in silico reconstruction of the temporal dimension and thus the investigation of the underlying molecular changes and circuitries. Several algorithms designed to reconstruct cell trajectories from single-cell data are available, each with distinct strengths and weaknesses[19-25].

Recent single-cell transcriptomic studies revealed that cell-cycle state and cell volume contribute to phenotypic and functional cell heterogeneity even in monoclonal cell lines[26,27]. This heterogeneity can obscure biological phenomena of interest[28,29]. For analysis of single-cell transcriptomic data, computational methods have been developed to reveal variability in cell-cycle state and cell volume; these methods use principal component analysis, random forests, LASSO, logistic regression, support vector machines, and latent variable models[26,28,30,31]. These methods leverage large numbers of previously annotated cell-cycle genes and are thus not transferrable to mass cytometry data analyses.

Here, we develop a combined experimental and computational method, called CellCycleTRACER, to quantify and correct cell-volume and cell-cycle effects in mass cytometry data. The application of CellCycleTRACER to measurements of three different cell lines over a 1-h TNFα stimulation time course reveals signaling features that had been otherwise confounded by cell-cycle and cell-volume effects.

## Results

### Cell-cycle and cell-volume effects measured by mass cytometry.
The impact of cell-cycle and cell-volume heterogeneity on mass cytometry data has not been addressed. We, therefore, set out to characterize how these factors influence commonly employed mass cytometry data analyses. To assess the effect of cell cycle, we exploited the simultaneous measurements of four cell-cycle markers recently identified by Behbehani et al.[32]: phosphorylated histone H3 (p-HH3), which peaks in the mitotic phase; phosphorylated retinoblastoma (p-RB), which monotonically increases from late G1 to M phase; cyclin B1, which increases from G2 to early M phase and rapidly diminishes during the late M phase; and 5-Iodo-2′-deoxyuridine (IdU), a thymidine analog incorporated during the S phase. We found that cell signaling as measured by protein phosphorylation strongly depended on the cell-cycle phase (Supplementary Note 1 and Supplementary Fig. 1). For example, a biaxial plot of phosphorylation of Ser241 on PDK1 vs. phosphorylation of Thr172 on AMPKα revealed that in G2 and M phases, phosphorylation levels were elevated (Fig. 1a). Consequently, the estimated Pearson correlation coefficient between these two markers appears to be high due to the G2 and M cells that inflate the correlation. Less dramatic

cell-cycle effects were also observed in published data[32] from a population of human T cells analyzed using a panel of immune-related cell-surface markers (Supplementary Fig. 2).

To assess the impact of cell volume, we had first to identify a marker that could be used to robustly quantify cell volume at a single-cell level. The ruthenium complex bis(2,2′-bipyridine)-4′-methyl-4-carboxybipyridine-ruthenium-N-succidimyl ester-bis (hexafluorophosphate) (ASCQ_Ru) stains proteins by covalently binding to amino groups[33] (Supplementary Fig. 3a). ASCQ_Ru can be used in mass cytometry to reliably measure cell volume, as demonstrated using confocal laser scanning microscopy and three-dimensional cell reconstruction (Supplementary Fig. 3b–d), provided that the cells are not under conditions where the total protein mass and volume become uncorrelated (e.g., under drastic changes in osmolarity). In mass cytometry, ASCQ_Ru is measured by the ion counts of seven ruthenium isotopes ($^{96}$Ru, $^{98}$Ru, $^{99}$Ru, $^{100}$Ru, $^{101}$Ru, $^{102}$Ru, $^{104}$Ru) that do not overlap with channels used for antibody measurements. Similarly to what we found for the cell cycle, the estimated correlation coefficients among phosphorylation markers were influenced by cell-volume heterogeneity (Fig. 1b, Supplementary Fig. 4). Given that the cell-cycle state and cell volume broadly confounded marker relationships, it was not surprising that analyses of mass cytometry data using standard statistical approaches, such as Pearson or Spearman correlations, or state-of-the-art computational methods, such as tSNE or DREMI, can result in misleading conclusions (Supplementary Figs. 1, 4).

### CellCycleTRACER normalizes cell-cycle and cell-volume effects.
Cell volume and cell cycle change in a continuous manner and should be corrected, or at least taken into account, accordingly. Therefore, we developed CellCycleTRACER, an algorithm for the analysis of single-cell mass cytometry data that enables correction for cell-cycle state and cell-volume heterogeneity. CellCycleTRACER is implemented as a simple and intuitive graphical user interface and can be applied to any mass cytometry data set. Its application requires that four channels be dedicated to the cell-cycle markers p-HH3, p-RB, cyclin B1, and IdU (see the section "Software and Data Availability").

CellCycleTRACER first exploits the ASCQ_Ru signal to transform raw marker counts into single-cell volume-relative intensities (Fig. 1c, Methods and Supplementary Note 2). After cell-volume correction, CellCycleTRACER uses data on the aforementioned four cell-cycle markers to classify cells into discrete cell-cycle phases and order them on a continuous path analogous to cell-cycle pseudotime (Fig. 1d, Methods and Supplementary Notes 3, 4). To automatically classify cells according to cell-cycle stage, CellCycleTRACER exploits a new machine learning approach that combines decision trees and Gaussian mixture models (Supplementary Figs. 5, 6, Methods and Supplementary Note 3); this approach reproduced manual gating procedures with 98.9% accuracy (Fig. 1d, Supplementary Fig. 7). Next, the single cells are ordered on a continuum that traces cell-cycle evolution based on a novel trajectory reconstruction technique (Fig. 1d, e). To achieve this, CellCycleTRACER exploits the prior cell-cycle phase assignment and identifies the optimal one-dimensional embedding of the four cell-cycle markers that preserves the known order of the cell-cycle phases by minimizing ordering violations (Fig. 1e, Supplementary Fig. 8, Methods and Supplementary Note 4). Finally, the cell-cycle trajectories of the measured markers are obtained by projecting single-cell measurements onto the pseudotime dimension (Fig. 1d, lower right).

Reconstructed cell-cycle trajectories of the four markers used for the pseudotime inference (p-HH3, p-RB, cyclin B1, and IdU)

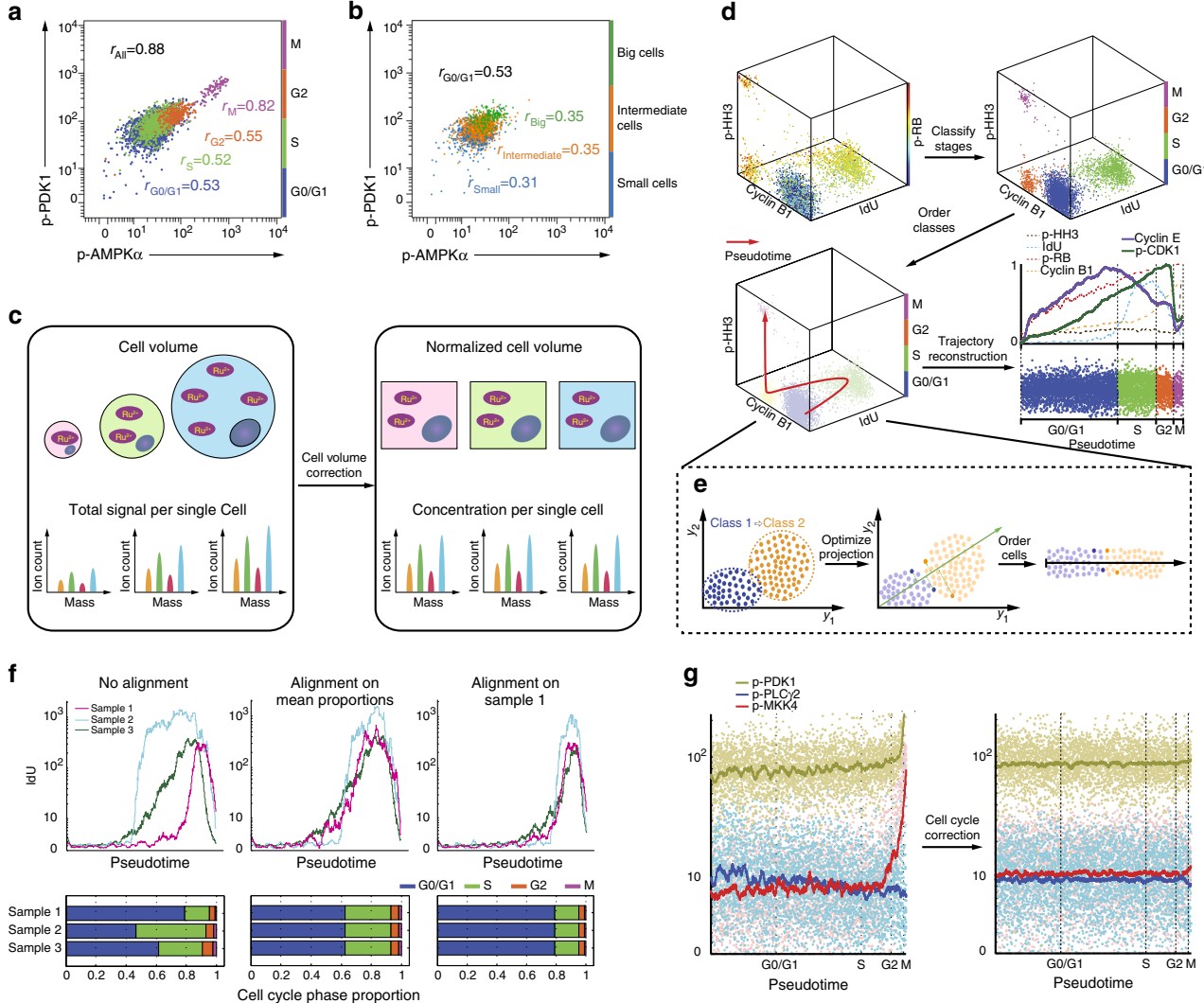

**Fig. 1** Cell-volume and cell-cycle biases in mass cytometry data and their corrections using CellCycleTRACER. **a** Biaxial plot of p-PDK1 (Ser241) vs. p-AMPKα (Thr172) in THP-1 cells, where pre-gated cell-cycle phases are indicated by different colors. Computation of Pearson correlation coefficients across cell-cycle phases indicates a strong cell-cycle bias. **b** Biaxial plot of p-PDK1 (Ser241) vs. p-AMPKα (Thr172) in G0/G1 phase THP-1 cells that were pre-gated by cell volume as indicated by different colors. Pearson correlation coefficients are indicative of the cell-volume bias. **c** Cell-volume correction using ASCQ_Ru measurements removes cell-volume variability and transforms raw counts of measured markers into relative concentrations at single-cell resolution. **d** Construction of cell-cycle pseudotime initiates with automatic classification of the cells into discrete cell-cycle phases using measurements of IdU, cyclin B1, p-HH3, and p-RB25. The optimal trajectory across phases is constructed by projecting the data in a one-dimensional embedding function analogous to cell-cycle pseudotime. Mean trajectories of all measured cell-cycle markers across the reconstructed pseudotime recapitulate known behavior. Markers used to construct the pseudotime (IdU, cyclin B1, p-HH3, and p-RB) are shown as dashed lines, additional cell-cycle markers used as validation (cyclin E and p-CDK1) are shown as solid lines. **e** Simplified example of the trajectory reconstruction technique. By exploiting prior information of the class labels for each cell and the order of the classes, the best embedding function is computed by selecting the one that optimally preserves the known ordering across all cells in the new subspace defined by the embedding. **f** CellCycleTRACER aligns cell-cycle pseudotime by equalizing cell-cycle phase duration across all analyzed samples. **g** CellCycleTRACER correction for cell-cycle redistributes the single cells independently of cell-cycle variation

agreed with their cell-cycle-dependent variation (Fig. 1d, dashed lines)[32,34]. Additionally, the pseudotime was validated by analyses of two independent cell-cycle markers, p-CDK1 (Tyr15) and cyclin E (Fig. 1d, solid lines). CellCycleTRACER results faithfully recapitulated prior biological knowledge. Phosphorylation of Tyr15 on CDK1 progressively increased during S and G2 phase, peaked at the G2/M transition and sharply decreased after the entry to M phase. Cyclin E progressively accumulated during the G1 phase and reached the maximum at the G1/S transition before being degraded during the S phase[35,36] (Fig. 1d, Supplementary Fig. 9). Furthermore, comparison with five state-of-the-art

trajectory and embedding reconstruction methods showed that these methods failed to reproduce biologically relevant orderings of the cell cycle (Supplementary Fig. 10). For example, Wanderlust[19] ordered the cells from G1→G2→S→M. Since Wanderlust works by first constructing $k$ l-nearest neighbor graphs in the four-dimensional space of the cell-cycle markers and assumes that changes in protein abundance levels are gradual in the trajectory, it traversed the data in the wrong order because the G1 cluster is closer to G2 than S due to the jump in IdU. The other methods tested resulted in different incorrect orderings. SCUBA[49] constructed a G2→S→G0/G1 trajectory and

incorporated the M phase cells in the other clusters; TSCAN[50] constructed a M→S/G2→G0/G1 trajectory by mixing together G2 and S cells; and Monocle ordered the data as G0/G1→M→G0/G1→G2→S, by ordering M phase cells in the middle of the G0/G1 cluster. Last, diffusion maps[48] yielded a non-linear, low-dimensional embedding of the data that did not capture the known ordering. Since these methods are unsupervised techniques, they reconstruct continuous trajectories of the given measurements with no additional label information. It is thus impossible to "force" these methods traverse the data in the known cell-cycle phase order. CellCycleTRACER, however, exploits the known order of the phases through a mathematically well-defined optimization routine and guarantees by design that the known ordering will be preserved in the inferred one-dimensional embedding.

Reconstructed cell-cycle trajectories of cell-surface markers from a population of human T cells[32] indicated a continuous increase across the cell cycle for many of the proteins, peaking at the M phase (Supplementary Fig. 11). CellCycleTRACER can also remove cell-cycle-related inter-sample variations (due, for example, to the use of different cell lines or of the same cell line at different stimulation time points) and enables unbiased multi-sample analyses by trajectory alignment. This is achieved using a subsampling strategy that equalizes the relative cell-cycle phase proportions either to the mean inter-sample proportions or to the proportions of a user-selected sample (Fig. 1f, Methods and Supplementary Note 5). Last, CellCycleTRACER can correct for cell-cycle-related intra-sample variations by dividing the ordered single-cell values by the normalized mean trajectory (Fig. 1g, Methods and Supplementary Note 6). The data set can be exported after any step of the pipeline, facilitating the use of various downstream data analysis approaches.

**Assessing CellCycleTRACER with TNFα stimulation data.** To test the performance of our method, we measured the abundances of 25 protein phosphorylation sites, three housekeeping proteins, and three phenotypical markers in conjunction with the cell-volume and cell-cycle markers (Supplementary Table 1) in HEK293T (embryonic kidney), MDA-MB-231 (breast cancer), and THP-1 (monocyte) cells that had been stimulated with TNFα for 0, 5, 10, 15, 30, and 60 min (Methods). Analyses of cell volume at the control time point (0 min) showed that MDA-MB-231 cells had on average the largest volume, followed by HEK293T and THP-1 cells (Fig. 2a). After cell-volume correction using CellCycleTRACER, the single-cell-volume distributions in the three cell lines perfectly aligned (Fig. 2a). Marker abundances were strongly influenced by the cell-volume correction. For example, the amount of phosphorylated MKK4 (Ser257/Thr261) in THP-1 cells was twofold lower compared to the amounts in the other two cell lines when uncorrected for volume biases; after the correction, the amounts were nearly identical in each of the cell lines (Fig. 2a). After cell-volume correction, the coefficients of variation of the measured markers were reduced, indicating that our method corrected for cell-volume-dependent variations (Fig. 2a, bottom).

We next analyzed the cell-cycle evolution of different phosphorylation markers in response to TNFα stimulation. This analysis exposed cell-cycle-specific phosphorylation responses to stimulation. For example, in THP-1 cells, phosphorylation of p38 (Thr180/Tyr182) in response to TNFα stimulation was twofold stronger in G2/M phases compared to G0/G1 phase (Fig. 2b, left). The cell-cycle dependency of p38 phosphorylation was confirmed by flow cytometry analysis where a similar fold change across the cell-cycle phases was observed (Supplementary Fig. 12). It was reported previously that TNFα induces histone H3

phosphorylation that peaks at 30 min post-stimulation; this contributes to chromatin remodeling and enhances accessibility of DNA to transcriptional factor NFκB[37]. Analysis of the TNFα-stimulated THP-1 data using CellCycleTRACER revealed that this effect was cell-cycle dependent, as levels of phosphorylated histone H3 (Ser28) in the build up to the S phase were twice as high as in early G0/G1 or G2 phases of the cell cycle (Fig. 2b, right). The application of CellCycleTRACER aligned the trajectories and removed the bias introduced by the cell-cycle stage (Supplementary Fig. 13).

We next assessed the performance of CellCycleTRACER by comparing data before and after cell-volume and cell-cycle correction. First, Pearson correlation, Spearman correlation, and DREMI (a mutual information based metric)[15] were used to quantify the relationship strength between measured markers in the unstimulated THP-1 cell data. As expected, after cell-volume and cell-cycle correction, Pearson correlation, Spearman correlation, and DREMI values for two cell-volume markers, ASCQ_$^{102}$Ru and ASCQ_$^{104}$Ru decreased significantly (Fig. 2c). For the signaling relationship between p-PDK1 and p-AMPKα, which is also affected by cell-cycle stage (Fig. 1a), Pearson correlation, Spearman correlation, and DREMI values were reduced from 0.88, 0.58, and 0.60 to 0.49, 0.34, and 0.34, respectively, upon application of CellCycleTRACER (Fig. 2c). Importantly, CellCycleTRACER correction had a smaller effect on the known direct signaling relationship of p-ERK (Thr202/p-Tyr-204) to p-p90RSK (Ser380), with a Pearson correlation slightly reduced from 0.77 to 0.69, Spearman correlation reduced from 0.75 to 0.67, and DREMI value reduced from 0.55 to 0.45 upon CellCycleTRACER application indicating that our method preserves real signaling relationships (Fig. 2c).

Second, we quantified the extent of cell-cycle-induced bias removed by CellCycleTRACER using an approach based on principal component analysis on a mixture of unstimulated ($t = 0$) and stimulated ($t = 15$ min) THP-1 cells (Fig. 2d). Specifically, after estimating the principal components of the data before and after cell-cycle correction, we fitted a linear model of the principal components on the cell-cycle-state index (i.e., G1, S, G2, and M phase) and the stimulation state and computed the variance explained by the fit ($R^2$) in all cases. Before correction, a large percentage of the variance in the first principal component was explained by the cell-cycle state; the effect was virtually eliminated by cell-cycle correction using CellCycleTRACER (Fig. 2d, left). Conversely, when the cell-cycle effect was eliminated, the increase of $R^2$ for components 2 and 3 indicates that a larger percentage of the variance in the data was explained by the stimulation.

Third, we used CellCycleTRACER to assess the impact of correction on signaling network reconstruction with DREMI. Pairwise DREMI analysis on unstimulated THP-1 cells indicated that before cell-volume and cell-cycle corrections, phosphorylation sites known to be elevated in the M phase, such as Ser529 on NFκB, Thr172 on AMPKα, Thr334 on MAPKAPK2, and Ser241 on PDK1, were clustered together (Fig. 2e, left). After CellCycleTRACER was used to correct for heterogeneity in cell volume and cell cycle, DREMI scores were reduced in general, but a clear pattern consistent with MAPK/ERK and the AKT pathway activation appeared (Fig. 2e, right). Without correction for cell-volume and cell-cycle effects, signaling networks reconstructed with the top 10 signaling relationships as identified with DREMI in unstimulated THP-1 cells did not agree completely with commonly accepted signaling knowledge (Fig. 2f, left), whereas with the correction canonical relationships were seen (Fig. 2f, right)[38–40]. Thus, pre-processing to correct mass cytometry data for cell-volume and cell-cycle heterogeneity is necessary for

accurate analyses of correlation and variance, mutual information-based signaling relationship analysis (performed here with DREMI), and signaling network reconstruction.

Finally, we assessed how the cell-volume and cell-cycle corrections performed with CellCycleTRACER influenced the

dimensionality reduction of a heterogeneous population of single cells with tSNE. Before the correction, the cell cycle confounded the separation of the cells in the tSNE plot (Fig. 2g, left), obscuring the cell line identities of the individual cells. The M phase cells from all analyzed cell lines were clustered, whereas

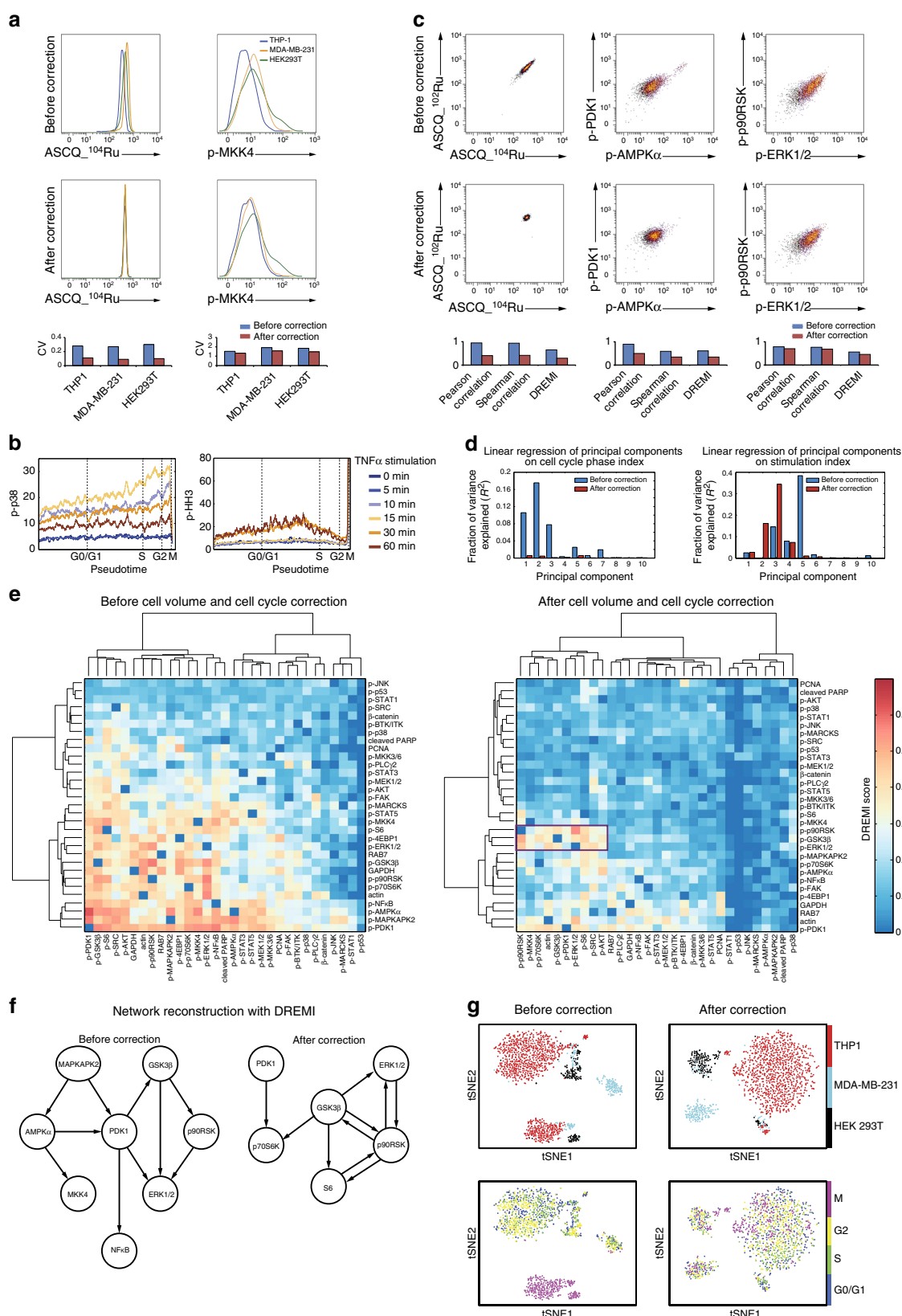

cells from all other cell-cycle phases were mixed, and MDA-MB-231 cells were separated into three clusters. After cell-cycle correction, three clusters corresponding to the three different cell lines were observed in the tSNE plot, and the cell-cycle origin of each cell in each cluster appeared random (Fig. 2g, right).

## Discussion

In summary, cell volume and cell cycle can confound downstream mass cytometry data analysis. The presented experimental and computational approach, which we call CellCycleTRACER, corrects for the influences of volume and cell-cycle phase on mass cytometry data. CellCycleTRACER is a supervised manifold learning method that, in contrast to existing methods, exploits the cell-cycle phase labels to guarantee that the known ordering will be preserved in inferred embedding. With CellCycleTRACER we provide the mass cytometry community with a method for supervised comprehensive analysis of cellular transitions. We expect that use of CellCycleTRACER will be particularly important when highly heterogeneous cell populations with deregulated cellular processes, as typically found in tumors, are analyzed.

## Methods

**Cell culture.** HEK293T, MDA-MB-231, and THP-1 cells were obtained from ATCC. HEK293T cells were cultured in Dulbecco's Modified Eagle's Medium (DMEM, D5671, Sigma) supplemented with 10% fetal bovine serum (FBS), 2 mM L-glutamine, 100 U/ml penicillin, and 100 μg/ml streptomycin. MDA-MB-231 and THP-1 cells were cultured in Leibovitz's L-15 Medium (11415064, Gibco) and RPMI-1640 Medium (52400025, Gibco), respectively, both supplemented with 10% FBS, 100 U/ml penicillin, and 100 μg/ml streptomycin. For passaging or harvesting, HEK293T and MDA-MB-231 cells were first detached by incubating with 1X TrypLE™ Express (Life Technologies) for 2 min at 37 °C.

**TNFα stimulation.** HEK293T, MDA-MB-231, and THP-1 cells were seeded in six-well plates at densities of 0.7 million cells, 0.5 million cells, and 1 million cells per well, respectively. After 2 days, cells were stimulated with TNFα (R&D Systems) at 10 ng/ml. Aliquots were collected for analysis at 0, 5, 10, 15, 30, and 60 min (stimulation was performed in reverse order to enable simultaneous harvesting of all conditions). At 20 min before harvesting, IdU was added to the medium at the final concentration of 10 μM. At 2 min before harvesting HEK293T and MDA-MB-231 media were replaced with 1X TrypLE to induce detachment. At the time of harvest, paraformaldehyde (PFA, Electron Microscopy Sciences) was added to the cell suspension at a final percentage of 1.6%, and samples were incubated at room temperature for 10 min. Crosslinked cells were washed twice with cell staining media (CSM, PBS with 0.5% BSA, 0.02% NaN₃). After removal of supernatant, ice-cold methanol was used to resuspend the cells, followed by a 10 min permeabilization on ice or long-term storage at −80 °C.

**Immunofluorescence and three-dimensional reconstruction.** CultureWell™ chambered coverglass wells (16-well, Thermo Fisher Scientific) were pre-coated by incubation with 10 μg/ml bovine plasma fibronectin (Thermo Fisher Scientific) at 37 °C for 1 h. MDA-MB-231 cells were then seeded at a density of 1500 cells per well. On the second day, 4% PFA was used to crosslink the cells at room temperature for 20 min. The slide was then washed with PBST (0.5% Tween 20 in PBS) three times, and cells were subsequently permeabilized for 5 min with 0.1% TritonX-100 (diluted in PBS) at room temperature. After washing with PBST three

times, cells were incubated in blocking buffer (10% goat serum diluted in PBST) for 30 min at room temperature. Primary antibody (anti-GAPDH, 6C5, Thermo Fisher Scientific, 1 μg/ml; anti-Rab7, D95F2, Cell Signaling Technology, 1:100; or anti-β-actin, D6A8, Cell Signaling Technology, 1:200) was added, and slides were incubated overnight at 4 °C. Secondary antibody (goat anti-mouse Alexa Fluor® 488, 1:200 or goat anti-rabbit Alexa Fluor® 555, 1:200), supplemented with Hoechst 33342 at a final concentration of 100 μg/ml, was applied, and slides were incubated for 1 h at room temperature. Antibodies were diluted in the blocking buffer. Slides were washed three times with PBST after each incubation step. For cell-volume analyses, cells were stained for total proteins with Alexa Fluor® 647 NHS ester (Thermo Fisher Scientific) for 10 min at a final concentration of 1 μg/ml. Slides were then mounted with ProLong Gold Antifade Reagent (Life Technologies) before imaging with a CLSM Leica SP5 microscope. Stacks were imaged every 0.5 μm, and the three-dimensional reconstruction and quantification of the total cell volume was performed with Imaris 7.7.2.

**Antibody conjugation.** Isotope-labeled antibodies were generated with MaxPAR antibody conjugation kit (Fluidigm) using the manufacturer's standard protocol. The antibody yield was determined based on absorbance at 280 nm. Candor PBS antibody stabilization solution (Candor Bioscience GmbH) was used to dilute antibodies for long-term storage at 4 °C.

**Barcoding and staining protocol.** Formalin-crosslinked and methanol-permeabilized cells were washed three times with CSM and once with PBS. Cells were incubated in PBS containing barcoding reagents (105Pd, 106Pd, 108Pd, 110Pd, 113In, 115In, and 139La) at a final concentration of 50 nM for 30 min at room temperature and then were washed three times with CSM[16]. Barcoded cells were pooled and stained with the metal-conjugated antibody mix (Supplementary Table 1) at room temperature for 1 h. The antibody mix was removed by washing cells three times with CSM and once with PBS. For DNA staining, iridium-containing intercalator (Fluidigm) was diluted in PBS with 1.6% PFA and incubated with the cells at 4 °C overnight. On the day before measurement, the intercalator solution was removed, and cells were washed with CSM, PBS, and doubly distilled H₂O sequentially. Total protein staining was performed with 25 μg/ml ASCQ_Ru (96631, Sigma) in 0.1 M NaHCO₃ solution for 10 min at room temperature. Cells were then washed with CSM, PBS, and doubly distilled H₂O sequentially. After the last wash, cells were resuspended in doubly distilled H₂O and filtered through a 70 μm strainer.

**Mass cytometry analysis.** EQ™ Four Element Calibration Beads (Fluidigm) were added to the cell suspension at a 1:10 ratio (v/v). Samples were analyzed on a Helios mass cytometer (Fluidigm). The manufacturer's standard operation procedures were used for acquisition at a rate of ~200 cells per second. After data acquisition, all .fcs files from the same barcoded sample were concatenated. Data were then normalized, and bead events were removed[41]. Doublets were removed, and cells were de-barcoded into their corresponding wells using a doublet-filtering scheme and single-cell deconvolution algorithm[42]. Subsequently, data were processed using Cytobank (http://www.cytobank.org/). Additional gating on the DNA channels (191Ir and 193Ir) was used to remove remained doublets, debris, and contaminating particulates. Manual gating was performed on IdU, cyclin B1, p-HH3, and p-RB to identify cell-cycle stages[32].

**CellCycleTRACER workflow.** CellCycleTRACER requires as an input measurements of the four cell-cycle markers (namely p-HH3, p-RB, cyclin B1, and IdU) as well as measurements of cell volume, ideally using the ASCQ_Ru markers.

**Data processing and cell-volume correction.** To determine cell volume at a single cell level, we initially experimented with three housekeeping proteins, namely GAPDH, actin, and RAB7 in HEK293T, MDA-MB-231, and THP-1 cells.

**Fig. 2** CellCycleTRACER corrects for cell-volume and cell-cycle heterogeneity enabling unbiased data visualization and downstream analysis. **a** Overlaid histograms reveal differential data observations before and after cell-volume correction. Bar charts show that cell-volume correction also reduces intra-sample variation as coefficients of variation of measured markers decrease. **b** Abundance of p-p38 (Thr180/Tyr182) and p-HH3 (Ser28) plotted on the cell-cycle pseudotime based on data from TNFα-stimulated THP-1 cells. Stimulation time points are indicated by different colors. **c** Biaxial plots show signaling relationships between measured markers before and after cell-volume and cell-cycle correction. Relationship strengths quantified by Pearson correlation, Spearman correlation, and DREMI are indicated in the corresponding barplots. **d** Principal component analysis of data originating from a mixed population of unstimulated (t = 0 min) and stimulated (t = 15 min) THP-1 cells. After computing the principal components of the data before and after cell-cycle correction, the variances explained by fitting a linear model of the principal components on the cell-cycle state index (left) and the stimulation state (right) were estimated, indicating removal of cell-cycle confounding effects. **e** Clustergrams of pairwise DREMI analyses of unstimulated THP-1 cells before and after cell-volume and cell-cycle corrections. After the removal of cell-volume and cell-cycle variability, DREMI scores of non-interactive pairs are reduced, and AKT and MAPK/ERK signaling pathways become apparent. **f** Network reconstruction using the top 10 DREMI scorers in unstimulated THP-1 cells before and after cell-volume and cell-cycle corrections. Network reconstructed after correction recapitulates key regulatory interactions in the AKT and MAPK/ERK pathways. **g** tSNE maps of THP-1, MDA-MB-231, and HEK293T cell lines before and after cell-volume and cell-cycle correction. Cell-cycle and cell-volume markers were not included in the tSNE analysis

Although all three proteins highly correlate with the total cell volume (Supplementary Fig. 14), single-cell measurements from a mixed population of three different cell lines revealed a large degree of cell-line-specific variability that surprisingly involved these housekeeping proteins (Supplementary Fig. 15). This indicated that these housekeeping proteins cannot be used for cell-volume correction when heterogeneous populations are analyzed.

ASCQ_Ru (Supplementary Fig. 3a) is conventionally used in electrophoresis for the determination of protein abundance and has been reported to outperform other staining reagents with high sensitivity and large linear dynamic range of protein binding. Taking advantage of its additional fluorescent property, we validated ASCQ_Ru as a precise cell volume indicator using three-dimensional reconstruction of confocal images (Supplementary Fig. 3b–d)). Measurements across the three cell lines showed reduced cell line variability in comparison to housekeeping protein measurements (Supplementary Fig. 16).

CellCycleTRACER corrects the data (uploaded as.fcs files) on ASCQ_Ru to enable correction for cell-volume heterogeneity. Let $j = 1, \ldots, m$ be the quantified protein marker in the $i = 1, \ldots, n$ single cell. Let $y_{i,j}$ denote the abundance of marker $j$ in cell $i$ in raw experimental data. At the same time, let $v = 1, \ldots, l$ denote the subset of the protein markers ($\{v\} \subset \{j\}$) that contain information on the total cell volume—in our case the ASCQ_Ru markers. During the cell-volume correction step, CellCycleTRACER first normalizes the raw cell-volume measurements $y_{i,v}$ by dividing each marker by its mean value:

$$y_{i,v}^{\text{norm}} = \frac{y_{i,v}}{\frac{1}{n}\sum_{i=1}^{n} y_{i,v}}.$$

The raw measurements of all $j = 1, \ldots, m$ markers are then corrected for cell-volume variations by dividing $y_{i,j}$ by the mean value of $y_{i,v}^{\text{norm}}$:

$$y_{i,j}^{\text{corr}} = \frac{y_{i,j}}{\frac{1}{l}\sum_{v=1}^{l} y_{i,v}^{\text{norm}}}. \tag{1}$$

Results of this process are shown in Supplementary Fig. 17. Comparison of measurements of the phosphorylated vs. total amount of proteins MEK1/2 and ERK1/2 before and after cell-volume correction indicated that the correction process was equally effective for both activated and total amounts of proteins (Supplementary Fig. 18). To avoid dividing by zero, CellCycleTRACER checks the data for zero values and, if found, substitutes zeros with the respective mean value. For more details on volume correction see Supplementary Note 2.

After cell-volume correction, selected channels of the raw measurements are transformed using the inverse hyperbolic sine function (asinh):

$$y_{i,j}^{\text{trans}} = \text{asinh}\left(y_{i,j}^{\text{corr}}\right) = \ln\left(\frac{y_{i,j}^{\text{corr}}}{c} + \sqrt{\left(\frac{y_{i,j}^{\text{corr}}}{c}\right)^2 + 1}\right), \tag{2}$$

where the constant $c$, commonly referred to as the cofactor, is set to 5 according to the CyTOF community's standard practice. Unlike the standard logarithmic function that is undetermined at zero values, asinh is linear around zero and becomes logarithmic beyond a threshold determined by the cofactor value. The overall effect of this transformation is to selectively compress large values, eliminating the typical long tails found in raw cytometry measurements. This results in a more symmetrical distribution that facilitates clustering and other machine learning analyses.

**Cell-cycle phase prediction**. After the volume correction, CellCycleTRACER classifies the single cells into discrete cell-cycle phases. To achieve this, we exploit the measurements of the four above-mentioned cell-cycle markers (IdU, p-HH3, cyclin B1, and p-RB) that are typically used in mass cytometry for manual cell-cycle gating. To eliminate possible biases introduced by variations in antibody concentrations and affinities, the data are standardized and set to have zero mean and unit variance. The prediction process is based on a hybrid approach that consists of two steps: (i) a classification step in which single-cell measurements of the four cell-cycle markers are given as input into a decision tree classifier to automatically predict the cell-cycle phase and (ii) a clustering step in which a Gaussian mixture model (GMM) is used to fit the data into clusters that represent the cell-cycle phases. The GMM is initialized using the predictions of the decision tree as prior knowledge.

Detailed descriptions of the implementation and performance on different data sets are given in Supplementary Note 3. In brief, we first used the four cell-cycle marker measurements from an experiment using THP-1 cells together with their class labels derived by manual gating (Supplementary Fig. 5a) to train a decision tree classifier. The resulting decision tree and the class proportions at the terminal nodes are shown in Supplementary Fig. 5b. We observed four pure terminal nodes, equal to the number of classes, indicating 100% classification accuracy in the training set. The order of the splits was identical to the order in the classification performed manually, indicating that the model faithfully captures the manual gating process. The decision tree accuracy in the independent test set was also 100%, meaning that all cells were correctly classified. After classification performance was validated, new experimental measurements were given as inputs

and were automatically classified. The results on a HEK293T test set are shown in Supplementary Fig. 7a.

Second, the measurements were clustered using a GMM where the number of components was set to four, equal to the number of cell-cycle phases in our model. The parameters of the GMMs (mean vectors, covariance matrices, and class proportions) were initialized using the decision tree predictions and iteratively refined until convergence using an expectation-maximization (EM) algorithm (Supplementary Fig. 6). After convergence, posterior probabilities of each of the four GMM components were computed, and the single cells were assigned to the component with the maximal posterior (results in Supplementary Fig. 7b).

This hybrid approach combines the intrinsic interpretability of decision trees, which enable extraction of a set of comprehensive if–else rules from the training data, with the probabilistic capabilities of the GMM framework. Specifically, decision trees are ideal for partitioning of a space using training data as prior knowledge, but they lack the notion of distribution and suffer from rigid boundaries. Since mass cytometry data are produced from different cell lines and possibly on different days, these data exhibit inter-experimental variability that makes the algorithm prone to misclassifying the points at the tails of the distribution. GMMs, on the other hand, allow flexible treatment of outliers. Shortcomings of GMMs—and other unsupervised clustering methods—are an inability to match the clusters to the known labels and no guarantee of convergence to the optimal solution. These limitations are especially acute when the classes are significantly imbalanced, as in the case of cell-cycle fractions which differ by an order of magnitude (e.g., G0/G1: 40–60%; M phase: 3–5% of the total cell population). By combining decision tree and GMM approaches we benefit from the advantages of decision trees to provide an initial guess close to the optimal solution and of GMMs to allow for a probabilistic interpretation of the class assignments. This refinement translates into better assignments for outliers and captures the classification uncertainty of cells transitioning between phases, a subtlety that is entirely missed by the decision tree.

**Trajectory reconstruction, alignment, and correction**. Cells progress along the cell cycle in a continuous way, gradually transitioning across consecutive phases whose boundaries are not always clearly defined, and exhibiting intra-phase variability (e.g., cells at early S and late S are drastically different). To better represent these pseudo-temporal fluctuations, we devised a method that reconstructs trajectories of biological cell-cycle time (*pseudotime*) from a population of unsynchronized single cells, ordering them according to cell-cycle progression. The details of the reconstruction method are given in Supplementary Note 4.

We assume that $n$ single cells are classified in four cell-cycle phases. Let $y_i$ denote a four-dimensional vector of cell-cycle marker abundances in each cell. We seek to construct a one-dimensional embedding function of the four-dimensional vector $y_i$, denoted as $f_\alpha(y)$, that represents pseudotime. One possible choice is to define $f_\alpha$ as a linear combination of $y_i$:

$$f_\alpha(y_i) = \sum_{j=1}^{4} \alpha_j y_{i,j},$$

where the coefficients $\alpha_j$ take values in $\mathbb{R}_{\geq 0}^4$. Under this formulation, our problem reduces to identifying a vector of coefficients $\alpha = (\alpha_1 \, \alpha_2 \, \alpha_3 \, \alpha_4)$ that optimally maps the cell-cycle marker measurements to pseudotime. Since the ordering of the discrete classes is known a priori (G1→S→G2→M), we follow an optimization process that aims to guarantee this ordering in the desired embedding by minimizing the difference across cells that belong to adjacent classes (see also Supplementary Fig. 8). More specifically, for all cells $i_p, i_q$ belonging to adjacent classes $p,q$, we estimate $\alpha$ such that $f_\alpha(y_{i_p}) < f_\alpha(y_{i_q})$, a constraint that translates into preserving the ordering in the embedding.

Collapsing the four-dimensional measurements into a lower-dimensional space may not result in fully separated clusters, with the implication that the ordering constraints might not be satisfied for all cells. To tackle this problem, we introduced slack variables into all constraints; these non-negative variables represent a degree of violation of the ordering constraint. We then minimize over a weighted sum of all slack variables, which leads to a mathematically well posed linear programming (LP) problem. Even though degenerate LPs can have multiple equivalent optima (convex set of optimal solutions), due to the presence of extrinsic and intrinsic variability in CyTOF data this does not occur in practice. Thus, the LP results in a single, optimal ordering. Since the solution time of the resulting LP grows substantially with the number of cells (Supplementary Fig. 19a) and can thus be computationally intensive, we randomly picked a fixed percentage of cells from each class and computed an optimal set of weights. A numerical investigation (Supplementary Fig. 19b) indicated that the solution of the LP, which yields the values of parameters $\alpha$, is robust to the sampling of cells even when a small percentage of cells is considered.

Once the values of parameters $\alpha$ are estimated, CellCycleTRACER visualizes the results by ordering the single cells based on their pseudotime values, resulting in single-cell trajectories for each marker. Additionally, CellCycleTRACER computes the mean trajectory of each marker by applying a mean filter on the single-cell trajectory, where the value for each cell is replaced by the mean of the neighboring cells in a sliding window of fixed size. Since different samples (e.g., different cell

lines) can exhibit strong variations in the relative duration of the cell-cycle phases, CellCycleTRACER permits multi-sample analysis by either aligning the relative cell-cycle phase proportions across individual samples to the mean vector of cell-cycle phase proportions or, alternatively, aligning them to one sample (e.g., one cell line) of interest (Fig. 1f, Supplementary Note 5). To remove the effect of the cell cycle on the marker measurements, CellCycleTRACER exploits the abovementioned mean trajectory, rescales it around 1 by dividing by the mean abundance of the marker and then divides the single-cell trajectory by the rescaled mean. This step removes cell-cycle-specific fluctuations and redistributes the single cells independently of cell-cycle variation (Fig. 1g, Supplementary Note 6). After every step of the analysis (e.g., cell-cycle classification, correction, alignment), data can be exported as .fcs files for further analysis.

**Implementation**. All methods were implemented using the Statistical and Optimization Toolboxes of MATLAB R2011b.

**Software availability**. CellCycleTRACER is implemented as a web application, accessible using the following link: https://www.zurich.ibm.com/compsysbio/publications.html.

**Data availability**. CyTOF data for the three cell lines at all stimulation time-points are available on Cytobank under project 1129.

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

## Acknowledgements

We would like to thank the Bodenmiller laboratory for support and fruitful discussions, the Pelkmans laboratory for sharing experimental materials, Dr Stéphane Chevrier for the help with flow cytometry analysis, Dr Vinko Tosevski and Dr Tess Brodie at the Mass Cytometry Facility, University of Zürich, and Dr José María Mateos Melero at the Center for Microscopy and Image Analysis, University of Zürich for support and trouble-shooting help. B.B.'s research is funded by an SNSF R'Equip grant, an SNSF Assistant Professorship grant (PP00P3-144874), by the European Research Council (ERC) under the European Union's Seventh Framework Program (FP/2007–2013)/ERC Grant Agreement n. 336921 and an NIH grant (UC4 DK108132). B.B. and M.R.M. are also funded by the SystemsX MetastasiX grant.

## Author contributions

X.–K.L. and B.B. conceived the study and experiments. X.-K.L. developed reagents and performed all experiments. X.-K.L., M.A.R., and M.R.M. performed data analysis. M.A.R.

and M.R.M. conceived the CellCycleTRACER algorithm. X.-K.L. and B.B. performed the biological analysis and interpretation. S.W. and M.L. conceived the optimization algorithm. B.B., M.A.R., X.-K.L and M.R.M. wrote the manuscript with input from all authors.

## Additional information

**Competing interests:** The authors declare no competing financial interests.

