## [Peer Review File · Nature Communications]

Editorial Note: This paper was previously reviewed at a journal not partaking in a Transparent Peer Review scheme. This Peer Review File contains the reviewer comments and author responses while at Nature Communications.

Reviewers' comments:

Reviewer #1 (Remarks to the Author):

Although I maintain concern about how widely used this method will be, most of my specific concerns have been adequately addressed. The exception being the one pertaining to the rebuttal document Q1-2 (supplementary figure 2). The authors show that CD4 was heavily varying along the cell cycle. We note that the figure shows that the distribution of CD4 expression is bimodal. (For this figure, please redraw so that the histograms can be independently viewed – as it is not possible to see the CD4 expression distribution of G0 cells) This is expected as CD4 is an extremely well known marker, that notably distinguishes CD4 T cells from CD8 T cells. It is extremely unlikely that CD4- T cells could up or down-regulate CD4 according to the cell cycle so much that they'd become CD4 positive or negative. The authors interpretation is thus extremely misleading. It is highly likely that cells that are proliferating are enriched for CD4 T cells, rather than "the cell cycle is confounding this study". It appears that Ki67 would have likely been enough to show an equivalent result (CD4 T cells show higher proliferation rates than the other cell types). Thus this example does not support the authors claim to have positively demonstrated that their approach is useful in immunological studies.

Reviewer #2 (Remarks to the Author):

In the manuscript "CellCycleTRACER – an Approach to Account for Cell Cycle and Volume in Mass Cytometry Data", now in revision at Nature Communication, the authors develop a new method for accounting for cell size and cell cycle stages in mass cytometry data.

I believe the authors addressed many of my points. However, there are two major aspects that have been addressed poorly and not added to the manuscript. The first point is the progression internally of each cell cycle stage and second is the accuracy in the transition between cell cycle stages. Please see below the specific questions.

1.

Q2-8: It is not clear how progression internally of each cell cycle stage is reflected in the trajectory. It would be important to see other cell cycle marker specific for the different cell cycle stages along the trajectory (these markers should not be used to build the trajectory).

R2-8: A good example to address this question is our used DNA marker. The abundance levels of DNA (191Ir) are constant throughout G0/G1, start to increase when the cells enter S phase, and peak at G2 at approximately double the amount observed in G0/G1 (Figure 10). Since DNA measurements were not used as input in the trajectory inference, this observation validates that the reconstructed trajectory indeed captured correctly the evolution of cells across the cell cycle. As mentioned above, due to some nonspecific binding of the used iridium intercalator, high levels of noise are observed for the DNA channel, which, however, do not affect the conclusions drawn here.

New Comment: This is an example for S phase but to my knowledge there is not quantifications in the other cell cycle stages, especially in G1 and G2. I believe this aspect is important and needs to be addressed also in the manuscript.

2.

Q2-7: Transitions between cell cycle stages are very important and trajectories should be able to recapitulate them accurately. Are transition states between cell cycle phase accurate and sharp? For example, in Fig.1g PCNA increase linearly over the trajectory while it should be absent in G1 and only increase at G1/S transition.

R2-7: The reviewer raises an important question. We agree that cell-cycle trajectories should be able to recapitulate major cell-cycle events accurately. Among the four cell-cycle markers we used in this study, p-RB smoothly increases during the cell cycle, cyclinB1 increases from S phase on, IdU incorporation starts at S phase, and phosphorylation on histone H3 increases sharply at M phase. Together, our approach confirmed the prior knowledge of these markers, and identified both gradual and sharp changes of abundance in the cell-cycle trajectory (see main text Fig. 1d), confirming the robustness of our method.

Previous research from different research groups using time-lapse imaging on multiple cell lines indicates that PCNA is present in cells in the G1 phase, that there is a gradual upregulation during S and/or G2 phases, and then a decrease at M phase^{20–22}. All these behaviors were recapitulated by analyzing manually gated cell-cycle phases (Figure 9) and by cell-cycle trajectory analysis (main text Fig. 1g). As such, our analysis confirmed prior knowledge of PCNA behavior through the cell cycle. Imaging-based studies^{20–22} also revealed nucleolar aggregation of PCNA during the S phase. However, since mass cytometry measures the total abundance of PCNA protein in a cell, this cellular event cannot be observed.

New question: I agree with the authors that the general progression through the cell cycle is maintained but I do not think my specific question has been addressed. What I would like to see is a protein or a phospho-state (not IdU that is used for the building of the trajectory) peaking specifically at transitions G1/S or S/G2 and have a quantification of how accurate the transitions are. I do not think CellCycleTRACER needs to be extremely accurate because it is not the purpose of the method but I believe that an evaluation and quantification of this feature in the manuscript is important.

We would like to thank the reviewers for their positive feedback. To address the remaining points raised by the reviewers, we have performed one additional experiment to validate our approach and have modified the main and supplementary text and figures accordingly. We hope that the reviewers will find our responses to their comments satisfactory.

Reviewers' comments:

Reviewer #1 (Remarks to the Author):

Although I maintain concern about how widely used this method will be, most of my specific concerns have been adequately addressed. The exception being the one pertaining to the rebuttal document Q1-2 (supplementary figure 2). The authors show that CD4 was heavily varying along the cell cycle. We note that the figure shows that the distribution of CD4 expression is bimodal. (For this figure, please redraw so that the histograms can be independently viewed – as it is not possible to see the CD4 expression distribution of G0 cells) This is expected as CD4 is an extremely well known marker that notably distinguishes CD4 T cells from CD8 T cells. It is extremely unlikely that CD4- T cells could up or down-regulate CD4 according to the cell cycle so much that they'd become CD4 positive or negative. The authors' interpretation is thus extremely misleading. It is highly likely that cells that are proliferating are enriched for CD4 T cells, rather than "the cell cycle is confounding this study". It appears that Ki67 would have likely been enough to show an equivalent result (CD4 T cells show higher proliferation rates than the other cell types). Thus this example does not support the authors claim to have positively demonstrated that their approach is useful in immunological studies.

Reply: We thank the reviewer for pointing out the misinterpretation of the results in the previous revision. Indeed, we agree that CD4 expression on CD4 T cells is not up- or downregulated during the cell cycle; rather, CD4 T cells are more proliferative than other T cell populations in this dataset. We did notice that the population of CD4-positive cells (the right peak of the bimodal histogram in Supplementary Fig. 2) indicates a progressive increase in CD4 expression as the cell cycle progresses. Similarly, the abundance of other evaluated CD molecules also varies with respect to the cell-cycle state. For example, the level of CD45RA is cell-cycle dependent, as previously reported by others¹. We understand that for typical immunological studies that characterize cell populations using surface marker levels, the confounding factor of the cell-cycle state would not have a drastic effect on population identifications. However, many studies focus on comparing the absolute expression levels of CD molecules and examine their up- or downregulation under certain conditions²⁻⁴. For these studies, it is important to be aware of the potential cell-cycle effect on the level of the CD markers (i.e., a treatment perturbs the cell

cycle will result in varied expression levels of CD molecules). In such scenarios, our approach to detect potential confounding factors and remove them from the data will be highly useful.

We have redrawn Supplementary Figure 2 to make all histograms visible, removed CD4 to avoid confusion, and rephrased our conclusions in the main text and in the legend of Supplementary Figure 2 so that readers are not misled.

Reviewer #2 (Remarks to the Author):

In the manuscript “CellCycleTRACER – an Approach to Account for Cell Cycle and Volume in Mass Cytometry Data”, now in revision at Nature Communication, the authors develop a new method for accounting for cell size and cell cycle stages in mass cytometry data.

I believe the authors addressed many of my points. However, there are two major aspects that have been addressed poorly and not added to the manuscript. The first point is the progression internally of each cell cycle stage and second is the accuracy in the transition between cell cycle stages. Please see below the specific questions.

Reply: We thank the reviewer for the positive feedback and the constructive comments regarding the internal progression of cell-cycle stages and the accuracy of our described cell-cycle phase transitions. Since both comments refer to the question how accurately our method recapitulates the actual cell cycle progression continuum, we address the comments in a single response below.

1. Q2-8: It is not clear how progression internally of each cell cycle stage is reflected in the trajectory. It would be important to see other cell cycle marker specific for the different cell cycle stages along the trajectory (these markers should not be used to build the trajectory).

R2-8: A good example to address this question is our used DNA marker. The abundance levels of DNA (191Ir) are constant throughout G0/G1, start to increase when the cells enter S phase, and peak at G2 at approximately double the amount observed in G0/G1 (Figure 10). Since DNA measurements were not used as input in the trajectory inference, this observation validates that the reconstructed trajectory indeed captured correctly the evolution of cells across the cell cycle. As mentioned above, due to some nonspecific binding of the used iridium intercalator, high levels of noise are observed for the DNA channel, which, however, do not affect the conclusions drawn here.

New Comment: This is an example for S phase but to my knowledge there is not quantifications in the other cell cycle stages, especially in G1 and G2. I believe this aspect is important and needs to be addressed also in the manuscript.

2. Q2-7: Transitions between cell cycle stages are very important and trajectories should be able to recapitulate them accurately. Are transition states between cell cycle phase accurate and sharp? For example, in Fig.1g PCNA increase linearly over the trajectory while it should be absent in G1 and only increase at G1/S transition.

R2-7: The reviewer raises an important question. We agree that cell-cycle trajectories should be able to recapitulate major cell-cycle events accurately. Among the four cell-cycle markers we used in this study, p-RB smoothly increases during the cell cycle, cyclinB1 increases from S phase on, IdU incorporation starts at S phase, and phosphorylation on histone H3 increases sharply at M phase. Together, our approach confirmed the prior knowledge of these markers, and identified both gradual and sharp changes of abundance in the cell-cycle trajectory (see main text Fig. 1d), confirming the robustness of our method.

Previous research from different research groups using time-lapse imaging on multiple cell lines indicates that PCNA is present in cells in the G1 phase, that there is a gradual upregulation during S and/or G2 phases, and then a decrease at M phase^{20–22}. All these behaviors were recapitulated by analyzing manually gated cell-cycle phases (Figure 9) and by cell-cycle trajectory analysis (main text Fig. 1g). As such, our analysis confirmed prior knowledge of PCNA behavior through the cell cycle. Imaging-based studies^{20–22} also revealed nucleolar aggregation of PCNA during the S phase. However, since mass cytometry measures the total abundance of PCNA protein in a cell, this cellular event cannot be observed.

New question: I agree with the authors that the general progression through the cell cycle is maintained but I do not think my specific question has been addressed. What I would like to see is a protein or a phospho-state (not IdU that is used for the building of the trajectory) peaking specifically at transitions G1/S or S/G2 and have a quantification of how accurate the transitions are. I do not think CellCycleTRACER needs to be extremely accurate because it is not the purpose of the method but I believe that an evaluation and quantification of this feature in the manuscript is important.

Reply: The reviewer made two important suggestions: First, the internal progression within each cell-cycle phase needs to be validated, and, second, the transitions at G1-S and G2-M checkpoints need to be accurately captured^{5,6}. As the reviewer correctly points out, although the trajectories of the four cell cycle markers (IdU, cyclin B1, pHH3, and pRB) accurately capture their expected behavior during the cell cycle, an independent validation using additional and well-established cell-cycle markers is important.

To address the reviewer's concerns, we analyzed two additional cell-cycle markers, namely cyclin E (peaks at the G1-S transition) and p-CDK1 (Tyr15) (peaks at G2-M transition). We analyzed the expression of these markers, together with the cell-cycle marker set used in the previous version of the manuscript, in THP-1 cells. We then reconstructed the cell cycle trajectory using IdU, cyclin B1, pHH3, and pRB with CellCycleTRACER. Although not used for

the cell-cycle reconstruction, CellCycleTRACER recapitulated the known regulations of cyclin E and p-CDK1 through the cell cycle^{7,8} (see figure below). Phosphorylation of CDK1 on Tyr15 progressively increased during S and G2, peaked at the G2-M transition, and was dephosphorylated once cells entered M phase (panel a). Cyclin E progressively increased during G1, peaked at the G1-S transition, and degraded during S phase (panel b). The inferred cell-cycle trajectories of these two well-known cell cycle markers demonstrate that our approach accurately captures S and G2 internal progression and the G2-M transition (p-CDK1) as well as the G1 phase progression and G1-S transition (cyclin E).

This new figure has been added to the revised manuscript as new Supplementary Figure 9, and Figure 1d has been adapted to display the new results. To avoid any further confusion regarding PCNA cell-cycle dynamics, we replaced the trajectory of PCNA with p-PDK1 in Figure 1g.

References

1. LaSalle, J. M. & Hafler, D. A. The coexpression of CD45RA and CD45RO isoforms on T cells during the S/G2/M stages of cell cycle. *Cell. Immunol.* **138**, 197–206 (1991).
2. Kobayashi, S. *et al.* Association of CD26 with CD45RA outside lipid rafts attenuates cord blood T-cell activation. *Blood* **103**, 1002–10 (2004).
3. Ehninger, A. *et al.* Distribution and levels of cell surface expression of CD33 and CD123 in acute myeloid leukemia. *Blood Cancer J.* **4**, e218 (2014).
4. Pollard, J. A. *et al.* Correlation of CD33 expression level with disease characteristics and response to gemtuzumab ozogamicin containing chemotherapy in childhood AML. *Blood* **119**, 3705–11 (2012).
5. Behbehani, G. K. in 105–124 (Humana Press, New York, NY, 2018). doi:10.1007/978-1-4939-7371-2_8
6. Behbehani, G. K., Bendall, S. C., Clutter, M. R., Fantl, W. J. & Nolan, G. P. Single-cell mass cytometry adapted to measurements of the cell cycle. *Cytometry. A* **81**, 552–66 (2012).
7. Bertoli, C., Skotheim, J. M. & de Bruin, R. A. M. Control of cell cycle transcription during G1 and S phases. *Nat. Rev. Mol. Cell Biol.* **14**, 518–528 (2013).
8. Castedo, M., Perfettini, J.-L., Roumier, T. & Kroemer, G. Cyclin-dependent kinase-1: linking apoptosis to cell cycle and mitotic catastrophe. *Cell Death Differ.* **9**, 1287–1293 (2002).

REVIEWERS' COMMENTS:

Reviewer #2 (Remarks to the Author):

In the manuscript “CellCycleTRACER – an Approach to Account for Cell Cycle and Volume in Mass Cytometry Data”, now in revision at Nature Communication, the authors develop a new method for accounting for cell size and cell cycle stages in mass cytometry data. This is an important aspect that needs to be taken into consideration in analysing such single-cell datasets.

I believe the authors addressed all my point by adding in the main text and in supplementary figures the new data. The work is solid and show strong reproducibility. The GUI allow different laboratory to use this method.